# Experiences of psychotropic medication use and decision-making for adults with intellectual disability: a multistakeholder qualitative study in the UK

Rory Sheehan  ,[1] Angela Hassiotis,[1] André Strydom,[2] Nicola Morant  [1]

[1]Division of Psychiatry, University College London, London, UK
[2]Department of Forensic and Neurodevelopmental Science, Institute of Psychiatry, Psychology and Neuroscience, London, UK

**Correspondence to**
Dr Rory Sheehan;
r.sheehan@ucl.ac.uk

## ABSTRACT

**Objectives** Understanding patient and carer perspectives is essential to improving the quality of medication prescribing. This study aimed to explore experiences of psychotropic medication use among people with intellectual disability (ID) and their carers, with a focus on how medication decisions are made.

**Design** Thematic analysis of data collected in individual semistructured interviews.

**Participants and setting** Fourteen adults with ID, 12 family carers and 12 paid carers were recruited from specialist psychiatry services, community groups, care providers and training organisations in the UK.

**Results** People with ID reported being highly compliant with psychotropic medication, based on a largely unquestioned view of medication as important and necessary, and belief in the authority of the psychiatrist. Though they sometimes experienced medication negatively, they were generally not aware of their right to be involved in medication decisions. Paid and family carers reported undertaking a number of medication-related activities. Their 'front-line' status and longevity of relationships meant that carers felt they possessed important forms of knowledge relevant to medication decisions. Both groups of carers valued decision-making in which they felt they had a voice and a genuine role. While some in each group described making joint decisions about medication with psychiatrists, lack of involvement was often described. This took three forms in participants' accounts: being uninformed of important facts, insufficiently included in discussions and lacking influence to shape decisions. Participants described efforts to democratise the decision-making process by gathering information, acting to disrupt perceived power asymmetries and attempting to prove their credibility as valid decision-making partners.

**Conclusions** Stakeholder involvement is a key element of medication optimisation that is not always experienced in decisions about psychotropic medication for people with ID. Forms of shared decision-making could be developed to promote collaboration and offer people with ID and their carers greater involvement in medication decisions.

### Strengths and limitations of this study

► This is the first multistakeholder study of patient, family carer and paid carer experiences of psychotropic medication use and the decision-making processes surrounding this for people with intellectual disability.
► Adaptations to qualitative methodology were made that allowed us to obtain meaningful data from people with intellectual disability.
► Using in-depth qualitative methods allowed us to develop a nuanced understanding of the relational and power dynamics underpinning decision-making about psychotropic medication.
► The views of prescribers and other health professionals are not included in this report.
► Those with limited or no verbal ability were not able to take part.

## INTRODUCTION

Up to 2% of the global population live with intellectual disability (ID), a lifelong condition characterised by significant deficits in cognitive and adaptive function with early onset.[1 2] A combination of biological, psychological, social and developmental factors contribute to a high rate of mental disorder in this group.[3] Recent evidence from epidemiological studies conducted across jurisdictions confirms that people with ID are often prescribed psychotropic medication, in many cases in the absence of a diagnosis for which it is indicated.[4–9] Psychotropic polypharmacy,[10–13] high doses[11] and increased susceptibility to adverse side-effects[14 15] are also significant concerns. Thus, people with ID are a key group in whom efforts to improve psychotropic prescribing are required. In England, a national programme, Stopping the Over-Medication of People with ID (STOMP), has

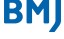

been established to reduce inappropriate use of psychotropic medication.[16] Co-produced with people with ID, the programme aims to raise awareness of the issue, develop resources for patients and carers, and act as a stimulus for practice change.[17]

Medication optimisation is a multifaceted approach to improving the use of prescribed medication with the aim of enhancing clinical outcomes, improving safety and reducing waste.[18] While deprescribing (reducing or discontinuing inappropriate medication) may be one element of optimisation, improving the quality of medication use requires more than a sole focus on quantitative measures. Understanding people's experience of medication and encouraging partnership between professionals and patients are also important components of successful medication optimisation.[18 19] As such, there are clear overlaps with several broader ideals and principles that are increasingly embedded in healthcare policies and clinical guidelines across health and social care internationally, including person-centred care, personalised medicine and shared decision-making (SDM). In relation to how decisions are reached about treatment options or courses of action, including use, choice and dose of medication, SDM seeks to replace traditional, paternalistic models with more collaborative approaches to treatment decisions where expertise and responsibility are owned jointly by the health professional and the patient.[20] The aims of SDM are congruent with longstanding UK government strategy to increase the inclusion and support the autonomy of people with ID in healthcare decisions and more generally.[21] As well as being an ethical ideal, evidence suggests that SDM is associated with a range of measurable benefits including improved understanding, patient satisfaction and trust.[22 23]

However, evidence indicates that people with ID may not routinely be placed at the centre of healthcare decisions[24] and carers of people with ID have reported that their views are not heard or that they are insufficiently involved by services.[25 26] The literature relating specifically to psychotropic medication in people with ID is less developed, though a small body of evidence shows that both people with ID and their carers often lack knowledge about psychotropic medication and experience few opportunities for involvement in medication decision-making.[27–30] It remains unclear how, and to what extent, the principles of SDM are applied in psychotropic medication decisions in contemporary UK settings. Additionally, how and between whom decisions are 'shared' in the clinical context of ID needs further exploration, as there are often multiple stakeholders in the form of family carers and those with paid caring responsibilities. In this study, we sought to explore the experiences and expectations of adults with ID and paid and family carers regarding psychotropic medication use, and how decisions about this are made with healthcare professionals.

## METHODS
### Participants and setting
People were eligible to participate if they were, adults (≥18 years) with ID who were currently prescribed psychotropic medication and were under the care of a specialist psychiatry of ID team, family carers of adults with ID who had been prescribed psychotropic medication, or paid carers who worked with adults with ID and who had experience of supporting people with psychotropic medication. Paid carers may have been employed in a variety of settings including residential homes, supported living projects or as peripatetic community support workers. Psychotropic medication was defined as any drug listed in the British National Formulary as being used for mental health disorders.[31]

The study was conducted in the south-east of England. Two methods of recruitment were used. In one, a leaflet advertising the research was offered to potential participants (people with ID, family carers, paid carers) by clinicians at appointments with specialist psychiatry of ID services within the National Health Service (NHS). These clinicians made a first assessment of eligibility to take part in the research. The other recruitment method included short presentations by researchers to community third-sector (ie, non-statutory), care provider and training organisations, with leaflets about the research also available. After hearing about the research, the contact details of those who showed an initial interest in taking part were passed to the research team, either directly from the person themselves or, with permission, via clinical staff. Potential participants were then contacted and eligibility was confirmed by liaison with people with ID and/or carers prior to interviews being held. The cognitive ability of potential participants with ID was not formally tested. Capacity to consent to taking part in the research was assessed immediately before the interview as part of the procedure of obtaining valid informed consent. This process was undertaken in accordance with the principles of the Mental Capacity Act[32] by a researcher with professional experience and training in assessing capacity. It was made clear to participants that their contribution was voluntary, that they could decline to take part without prejudice, and they may end an interview at any time. Written consent was received from all participants before interviews were conducted. Purposive sampling was used to select participants with a range of characteristics that may be related to medication views and experiences. For people with ID this included age, gender, ethnic group, indication for psychotropic medication and medication class; for family carers, age, gender, ethnic group, degree of ID in their relative, indication for and class of medication; and for paid carers, age, gender, ethnic group, duration working with people with ID, and seniority.

People with ID and family carers were given a £20 shopping voucher as a token of appreciation for donating time to the study. Paid carers were provided with a certificate thanking them for their contribution.

## Data collection

Baseline demographic and descriptive data were collected by participant report; we did not cross-check these with other sources of information. Qualitative data were collected in audio-recorded individual in-depth semistructured interviews conducted by the first author, who is a psychiatrist and clinician researcher with experience of working with people with ID and an academic interest in medication use. He did not have any other contact with participants. All interviews were conducted face-to-face. Participants were able to bring other people to their interview, if they wished, and interviews were held at a time and place preferred by participants. A topic guide with open-ended questions was developed and used to provide a broad structure to the interviews while allowing points of interest to be pursued as they arose. Interview topics included, people's experiences of using psychotropic medication, discussions medication with health professionals and how decisions about medication are made (see online supplementary material). Paid carers reported experiences and attitudes formed from supporting several different people. We adopted a flexible approach to interviews with people with ID in order to facilitate their involvement, including adapting the depth of questioning as appropriate to their ability.[33] All study materials for people with ID were available in 'easy-read' format and laminated picture cards were used (where appropriate) as prompts and to orientate interviewees. Checking and summarising content throughout the interviews gave opportunity for clarification and elaboration. Reflective field notes were made to supplement the transcripts and assist with reflexive practice and data analysis.

## Analysis

Descriptive data were summarised and tabulated. Audio-recorded interviews were transcribed verbatim by the first author, anonymised, and the transcripts checked for accuracy. As a research team we are interested in medication optimisation for people with ID and in how shared decision-making processes can impact this. Given the relative lack of literature in the field, thematic analysis was used with an inductive orientation in which themes were derived from the data.[34] Transcripts from each group of participants were analysed concurrently to build a unifying coding frame that was developed in an iterative process as additional transcripts were analysed. Independent coding of a subset of six transcripts by members of the research team early in the analytic process, regular discussion of emerging themes and the conceptual coherence of the findings, and reflexive memos were used to enhance integrity of the analysis. NVivo V.12 qualitative data analysis software (QSR International Pty, 2018) was used to manage the data and facilitate the analytic processes.

## Patient and public involvement

The development of the recruitment strategy, and the design of participant materials and the interview topic guide were informed by discussions with a consultation group consisting of people with ID employed for this work, some of whom had lived experience of mental illness, psychotropic medication use and contact with mental health services. The group will assist with future targeted dissemination activities to the participants with ID, their families and prescribers.

## RESULTS

### Sample

Thirty-eight people (14 adults with ID; 12 family carers; 12 paid carers) were recruited between December 2017 and May 2018 (table 1). Twenty-nine were recruited from clinical services and nine from third-sector organisations. Eighteen interviews were completed at peoples homes (10 people with ID; 8 family carers), 12 (all paid carers) at their place of work, 7 (3 people with ID; 4 family carers) at a university, and 1 (person with ID) at a community centre. Seven participants with ID preferred to have a companion with them in the interview (in six cases this was a relative, in one case a professional advocate).

Participants with ID reported having been diagnosed with a range of psychiatric disorders and most had been prescribed psychotropic medication for many years and in some cases for decades. None of those who participated were under a legal framework of care (eg, Community Treatment Order or Guardianship Order).

### Thematic analysis

We developed three major themes in our analysis of the data, and present these in each subsection below. The first theme, medication beliefs and experience, describes the meanings that people give to psychotropic medication, and how these can develop over time. The second theme, carer role, draws mainly on the interviews with paid and family carers to describe how the carer identity is constructed and how caring activities are performed. Together, these themes provide context to the third major theme about decisional processes, in which the lived experiences of different stakeholders in the medication decision-making process are explored, including the dynamics and struggles that sometimes characterised interactions with prescribers. Throughout the analysis we aim to provide a sense of the data by using quotes from anonymised participants who were given a number prefixed with ID (person with ID), FC (family carer) or PC (paid carer).

### Medication beliefs and experience: acceptance and ambivalence

We developed this theme predominantly from interviews with people with ID and family carers as we found that paid carers were generally more hesitant in offering their personal opinions about medication. In this theme, passive compliance of the person with ID emerged, founded on relatively limited understanding of medication, yet a strong sense of faith in medication and trust

| Table 1 | Sample characteristics | | |
|---|---|---|---|
| | **People with ID (n=14)** | **Family carers (n=12)** | **Paid carers (n=12)** |
| Mean age (SD, range) | 46.1 years (12.9, 25–68) | 62.7 years (10.5, 42–80) | 39.4 years (9.5, 24–55) |
| Sex (M:F) | 9:5 | 3:9 | 6:6 |
| Ethnic group | White (n=8)<br>Black (n=2)<br>Asian (n=3)<br>Other/mixed (n=1) | White (n=8)<br>Black (n=1)<br>Asian (n=3)<br>Other/mixed (n=0) | White (n=7)<br>Black (n=3)<br>Asian (n=2)<br>Other (n=0) |
| Degree of ID* | Mild (n=12)<br>Moderate (n=2) | Mild (n=6)<br>Moderate (n=4)<br>Severe-profound (n=2) | N/A† |
| Relationship to person with ID/professional title | N/A | Parent (n=10)<br>Other relative (n=2) | Support worker (n=8)<br>Managerial responsibility (n=4) |
| Mean time working with people with ID (SD, range) | N/A | N/A | 9.4 years (9.0, 0.5–25) |
| Current living arrangements | Independent (n=3)<br>With family (n=5)<br>Shared supported living (n=6) | With family member with ID (n=9)<br>Separately from family member with ID (n=3) | N/A‡ |
| Self-reported psychiatric diagnosis*§ | Severe mental illness† (n=6)<br>Depression (n=6)<br>Anxiety disorder (n=5)<br>Other (n=2) | Severe mental illness† (n=4)<br>Depression (n=4)<br>Anxiety disorder (n=6)<br>Other (n=0) | N/A‡ |
| Autism* | n=3 | n=5 | N/A‡ |
| Prescribed medication by group*§ | Antipsychotic (n=9)<br>Mood stabiliser (n=3)<br>Antidepressant (n=9)<br>Other (n=3) | Antipsychotic (n=10)<br>Mood stabiliser (n=2)<br>Antidepressant (n=9)<br>Other (n=4) | N/A‡ |
| Mean duration of psychotropic use (SD, range)* | 16.8 years (14.0, 3–50) | 13.6 years (8.0, 1–27) | N/A‡ |
| Mean interview duration (SD, range) | 24 min (9.0, 11–38) | 38 min (10.9, 19–55) | 47 min (11.9, 31–73) |

*Information provided by family carers relates to the person with ID they cared for.
†Severe mental illness includes schizophrenia spectrum disorders and bipolar affective disorder.
‡Data for paid carers were not collected as each paid carer worked with more than one individual with ID.
§Cell total exceeds the number in each group as people were able to report more than one diagnosis and may have been prescribed medication from more than one psychotropic class.
ID, intellectual disability; N/A, not applicable.

in the doctor. For family carers psychotropic medication was an emotive topic and many were ambivalent about its use. A minority of paid carers expressed concerns about inappropriate psychotropic use.

People with ID tended to focus on the tangible aspects of psychotropic medication (the taste, colour and size of tablets) and the set of 'rules' that constituted their current medication routine, for example, 'I take [the tablets] at night-time, the little mauve ones, my big yellow ones, and my little white sleeping tablet' (ID05). There was a tacit belief in medication as important and necessary, even though in many cases understanding of the indication for medication and its potential effects was limited. Most people with ID characterised medication benefits in vague or generic terms (eg, '[medication] gets me better' (ID01); 'it's helpful … for my health' (ID09);

'keeps me steady' (ID13)), while describing adverse side-effects using more immediate and vivid language (the most commonly mentioned were sedation, weight gain and movement side-effects):

My speech got slurred … really terrible and slurred. I just couldn't get the words out. (ID07)

I felt groggy … like I feel like a cabbage sometimes. (ID08)

The perceived consequences of not taking medication were often described as frightening and unpredictable and included being out-of-control or 'a danger' (ID10). Some feared they would 'probably end up back in hospital' (ID13) if they stopped medication, experiences of which (in those who had previous admissions)

were universally negative and acted as a strong motivator to keep well, which people equated with medication compliance. Although a minority of people with ID did express more critical views about medication or declared that they did not like taking it, none seriously questioned its use or believed there was an alternative:

> I don't want to take it… I don't like taking it, but I have to. (ID04)

> I don't like taking medication at the best of times, but I know I've got to take it. (ID10)

Given the length of time that most family carers had been managing medication (average >13 years), they tended to describe their experience as a journey and their narrative was often recounted with a strong emotional overlay. Many recalled that medication was first prescribed during a mental health crisis. In these difficult and stressful circumstances, which sometimes impacted their own mental health, family carers could find it difficult to make a confident decision about medication; the imperative to act being set against a fear of psychotropic drugs and their possible side-effects:

> In the beginning I was terrified about medication, the side-effects and everything. And also her [daughter's] condition… It's a really dangerous medication… I read lots of information and went on the internet, and it said lots about side-effects… But I didn't have any way out… I was really worried and couldn't make the decision. (FC08)

Initial reticence was often overcome when the beneficial effects of medication were observed and family carers could undergo quite major shifts in attitude:

> I'd always been quite resistant [to medication] because I'd heard about chemical coshes and all that stuff… I thought '[son] doesn't need a psychotropic' … but he went onto a very low dose and it noticeably helped… Now I'm at a stage of the psychiatrist thinking we should reduce the dose, and I'm really resistant to that because it feels so helpful. (FC02)

Others' longer-term experience of medication was less favourable. In these cases medication was variously described as ineffective, only temporarily effective (the positive effects 'wearing off' (FC01) over time was a common complaint), or blighted by adverse physical side-effects. The potential of psychotropic medication to dull people's cognitive faculties was expressed in various terms (eg, '[relative] was almost like a dead person … the drugs [meant] she was moving away from us … becoming a non-person' (FC12); 'they have this vacant kind of look … staring into the horizon' (PC01) 'a sledge hammer treatment' (PC07)). Fears about psychotropic medication were occasionally juxtaposed against the sensitivity and exceptionality of the person with ID:

> Sometimes I don't think these tablets are for people with autism and learning disabilities at all, you know?

> That's not the answer … if there's no cure, why are you giving all this medication? (FC03)

Some carers spoke of witnessing multiple medication changes and had come to view medication with scepticism, as unpredictable ('like taking pot luck' (FC09)) or even an 'experiment' (FC08 and FC12). Other concerns about medication included medication being used too readily ('[the doctors are] very quick to put them on but very slow to take them off' (FC06)); the absence of alternative, psychosocial interventions which were often considered more appropriate but unavailable due to resource constraints ('other things can cost money … so sometimes it's a control medication' (PC06)). Considering these concerns, for many carers psychotropic medication use was an ongoing source of tension and unease:

> I'm not happy with medication… The prescription is easy to write out … but medication might not be for [son] at all, for what's wrong with him, and they're writing out prescriptions all the time… He's got no other support around these issues … it's always just medication … not enough, err, not enough maybe talking therapy… I think there should be more done than there is. (FC03)

> Hopefully [relative will need] less medication in the future… I'm worried about the side-effects but also that she will become unwell if she stops [medication] … it's difficult, I don't know what will happen. There could be many problems. (FC07)

### Carer role: the 'front-line people'

In describing their roles in caring for a person with ID, both paid and family carers placed substantial importance on knowing and being close to the person, and the privilege that this gave them in evaluating their well-being. Carers also spoke of their role as advocates, ensuring that processes are centred around the person with ID and their interests are upheld.

In relation to psychotropic medication, in addition to practical, daily tasks such as collecting, storing and giving medication to the person with ID, both family and paid carers explained their 'integral' (PC02) role in monitoring and managing people's health. Carers described themselves as 'the front-line people', (PC01) a unique position which gave them intimate knowledge of the person with ID and was contrasted with 'short and limited' (PC05) meetings with medical professionals. Knowing the person with ID closely and over time was seen as important in view of the range of problems that were described among the group they supported (including physical illness, developmental disabilities, mental illness and/or behavioural problems). Given this complexity, carers perceived value in their ability to interpret subtle signs and to 'build up a picture of that person and how medication interacts with them' (PC02). Family carers, in particular, described an intuitive sense of 'knowing' the needs of their relative:

I've always had to deal with [son] not being verbal and not being able to tell me, so I had to read him by body language all through his life. I'm aware of the signs… I know if he has an infection in his nose, in his ears. I know if he has a headache … if he's not OK… I already know. (FC04)

Carers often took a 'gatekeeping' role in determining when to seek professional advice, and in mediating interactions between the doctor and the person with ID thereafter. Family and paid carers diverged slightly in how they positioned themselves during medical appointments. Family carers described taking a more direct approach in speaking with the doctor and acting on behalf of their relative, including, for example, one mother who attended appointments with the psychiatrist while her son waited outside the room. Paid carers, meanwhile, framed their input as 'empowering' (PC09) and facilitating the person with ID to speak for themselves, so that 'if there's something the service user wants to say, I can make sure it happens' (PC04) while preferring to take more of a 'back seat' (PC06).

Several carers spoke about a process of 'translating' (PC09) information between the doctor and the person with ID, again drawing on their familiarity of the person with ID in order to relay information in an individualised and more understandable way. This role often incorporated 'preparing the service user for the appointment and explaining in a very clear way what might happen' (PC04) and afterwards, reflecting with and educating the person with ID after the appointment:

[My relative] usually says [to the psychiatrist] 'it's best if you explain this to my mum or sister because they're good at explaining it to me'. (FC08)

I always get questioned by my clients 'What's this pill? What's that pill?' What I've done for my key clients is I've made a list of all the medication, and I did it in easy read …. and I've got a table of what they do with picture … if they ever ask me what happens, I just show them and go through it with them… I will stick it up on the fridge to familiarise people with it. (PC05)

In summary, carers viewed their role with respect to medication as both broad in scope and vital to the life of the person they supported:

I understand that sometimes I come across overbearing, nosey, and always getting involved … but I do believe, and this is a firm belief, if I was not behind [son] and asking for him, demanding for him … he would be in a worse place now, mentally… If he didn't have me he would definitely be worse off in all sorts. (FC09)

### Decisional processes relating to psychotropic medication

In this section we describe the forms of involvement that people with ID, their family carers, and paid carers experienced and desired in medication decisions, and their feelings and responses when these differed from the decisional processes they experienced.

### Power dynamics

There was a common assumption across stakeholder groups that the psychiatric appointment was the nexus of medication decisions and that the psychiatrist has the 'ultimate power' (FC02) and 'final say' (PC08) in medication decision-making. Interviewees did not express a desire to challenge this, viewing the psychiatrist as 'the expert' (FC11) who 'knows best' (ID10) and 'does the best for everyone who's sick' (FC07). In cases where people did not share the psychiatrist's opinion on medication, they relatively quickly deferred ('the medical profession probably know better…. I come on-board' (PC06)) and would not act alone to change medication:

I wouldn't [change medication] because then if anything happened I'd be the one to blame. It says in the leaflet 'do not stop medication unless you speak to your doctor' … sometimes I feel like doing it and I think to myself, 'no, I'll leave it and talk to [the psychiatrist] first' … they know better than we do. (FC03)

For many with ID the authority of the doctor was seen to be absolute and left little room for their own agency. Based on their lived experience, medication decisions were a part of life over which could exert little influence:

I have to take my medication, I ain't got no choice… It's the doctor's orders to keep on the medication … there's not a lot you can do about it. (ID11)

It's the doctor's decision [about medication] … it's up to them. (ID01)

Some people with ID were satisfied with the psychiatrist assuming control over medication decisions:

Doctors should make the decisions about medicine … they have more experience … [I prefer to] leave it to the doctor. (ID14)

However others (generally those with more mild ID) wanted to be involved in the process (eg, 'Explain what [the medication] is supposed to do… Tell me what's going on!' (ID06)). Congruent with these wishes, there were some descriptions of shared medication decisions. One woman with ID, for example, described how she had jointly reached a decision about reducing her medication, explaining that '[it was] my idea … and theirs [the doctors'] too' (ID04).

The desire of both paid and family carers to be involved in medication discussions and decisions was more obvious and evident through their depictions of both positive and negative experiences of medication decision-making across time and between clinicians. Positive experiences of medication decision-making were described as collaborations, 'partnerships' (FC02 and PC02) and 'negotiations' (PC08) and participants often made reference to having a good working relationship with the psychiatrist.

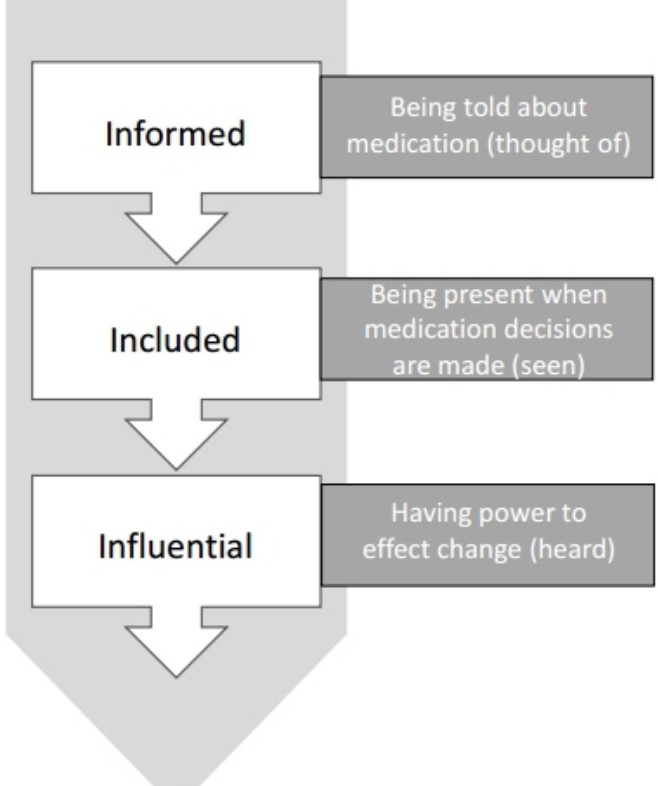

**Figure 1** Elements of involvement in medication decisions described by participants.

In these accounts, people valued 'open discussion' (PC09), being given 'time to talk' (FC10), invited to give their opinion, and being 'welcomed' (PC12) and 'taken seriously' (FC02) when doing so:

> It's been a really good partnership trying to get [service user] on the right medication… It's worked really well… I went along to see the psychiatrist, spoke to him about my concerns … and then he very quickly sent appointments through to see them. And I thought, 'wow, he listened, took it on board, called those people in, reviewed their medication'… The psychiatrists have been very tolerant, very patient and have listened to what we've been saying… So it can work. (PC02)

> A lot of doctors are open to discuss … they ask the [patient] and they ask me … and they listen. (PC06)

> [The doctor] was utterly supportive [and] took seriously what I'd said, so I trusted her… She suggested medication … it was made very clear to me what the long-term side-effects are… I wanted to give it a try, see how it goes. [I felt] no pressure… I think the professionals are very good at consulting. (FC02)

Conversely, being excluded from decisions about medication could take an emotional toll, especially on family carers who described feeling 'annoyed' (FC05), 'frustrated' (FC04 and FC08), 'angry' (FC12 and FC08) or isolated:

> It's always a bad experience when you're not involved… I wasn't in control of anything really, and there was no-one out there I could turn to. (FC11)

> It's been extremely stressful… When you find out somebody's been fiddling [with medication] behind your back and you haven't known about it. (FC05)

### Efforts to democratise medication decisions

From respondents' accounts of how medication decisions were made, we identified three related elements of decision-making. These were being informed, being included and having influence (figure 1). In any one of these processes, patients and carers could find themselves marginalised. Many paid and family carers, and a smaller number of respondents with ID, described making efforts to change the dynamics of medication decisions with strategies aimed at democratising each of these elements.

A pre-requisite to involvement in the decision-making process was to be informed about medication, yet several people with ID could not recall that medication was ever spoken about by their doctor ('I don't think [the psychiatrist] talks about medication… I ain't got a clue' (ID02)). These experiences reinforced a sense of powerlessness as medication decisions were perceived to 'just happen' (ID01). Both paid and family carers reported lacking information ('hardly ever told when people switch medication' (PC09))and sometimes 'not knowing what's going on' (FC05). Paid carers, particularly those working in larger organisations in which numerous people with ID were supported, worried that being 'out of the loop' (PC12) left them 'ill-equipped and dangerously exposed' (PC11), at once responsible for medication administration and monitoring yet without vital information of drug changes, doses or effects.

In response, both family and paid carers, and occasionally people with ID, had made attempts to improve their knowledge about medication (and alternative treatments) by seeking information independently from a variety of sources, including medication leaflets, television, internet, news media, carer networks, colleagues and formal training courses. People with ID were often reliant on carers to help them gain further information:

> My sister can come, we can look up what [the medication's] supposed to do so at least I get a better picture. (ID06)

Acquiring knowledge was reported by participants to improve their confidence and go some way to meet and respond to the technical expertise of the psychiatrist. Many people with ID, and some carers, however, could struggle with accessing appropriate information and were left in a relatively less powerful position as a result. None of the participants mentioned having used accessible medication information.

> Because I've got the learning difficulties, I'm not able to understand a lot… I'm not very good with a lot of the terms and conditions on there. It's really hard for

me to read one of those [medication information] leaflets… I don't know much about it so I can't say yes and I can't say no. (ID10)

Me myself is not very good in asking questions or understanding everything, so I just leave it… I can't go on the internet… I'm not very good in reading and writing, I don't understand everything, so that's why I don't bother. (FC07)

Respondents in all groups had experience of being nominally present when medication decisions were made but not *included* in discussions in a meaningful sense, and reported having little to no opportunity to voice their concerns:

They said 'you will be going on an anti-depressant.' I didn't know the name, then it all went cold …. the next thing I knew it was in my blister pack and I've been taking it ever since. (ID06)

I don't think my opinion was asked… I was in the review but I wasn't asked the big questions about treatment. (PC10)

Family and paid carers spoke of trying to shape the discourse in conversations with the psychiatrist and needing to have confidence to challenge their authority in order to ensure their views were heard. One relative described her assertive approach as 'not muck[ing] about… If I think the doctor's wrong, I tell 'em, just like that' (FC01). Sometimes a dramatic 'bust up' (FC09) or 'battle' (FC12) with the clinical team was considered necessary and could 'reset' the interaction in favour of a greater role for the family carer in medication decisions. At other times tenacity and 'pushing to be involved' (PC09) spoke of ongoing effort to develop and maintain involvement:

I always have to be chasing. I'm still chasing now… It shouldn't be like that, but that's the way it works… I think [the doctors] respect me more after, I kind of, put my foot down. (FC04)

Paid carers tended to avoid overt conflict. Instead they often relied on their accumulated knowledge of the healthcare system to navigate to a position where they stood the greatest chance of being heard. One paid carer described the strategy involved in arranging an appointment with the psychiatrist:

I'll have to write [to the psychiatrist] and copy in the GP… I'll have to be quite forceful about it. And then I'll actually ring [the psychiatrist] and I'll follow it up with an e-mail… We can ring the learning disability [team] secretary because we've got a very good relationship with her… I will actually sometimes say to her, 'it's quite a complex case this is, it's probably worth us seeing the consultant'. (PC08)

The final element to being involved that was described by respondents was the ability to *influence* decisions about medication. This constituted moving beyond

merely exchanging information to becoming a meaningful collaboration partner, whose opinions were heard and shaped decisions. Although there were clear instances where this had been achieved, all three stakeholder groups described situations in which this had not happened. Some also described strategies they had used in attempts to increase their decisional influence.

The minority of people with ID who had attempted to assert themselves were generally not successful in gaining the greater involvement and influence they wanted. In response to questioning their medication, some people with ID described receiving evasive answers that served solely to reinforce the importance of taking medication as directed:

I just get ignored, I feel like I'm getting ignored … when I say something about [medication], it's basically 'you just have to take the medication'. (ID08)

Sometimes I do [talk to the doctor about medication] but they tend to, like, they say 'we can't really say nothing because you've got to take it' and they don't really say why. (ID10)

One described having recruited a carer to advocate on their behalf but it was more common for people with ID to quickly acquiesce:

I don't get heard out properly… [The doctor says] 'Is [the medication] keeping you right?' and I just say 'yeah', but I don't think it is. But I don't want to argue. I don't want to argue with them so I just say 'yeah, it works on me'… I've asked [the psychiatrist] before to [change medication] but she wouldn't let me so I just let [the psychiatrist] get on with it… I just don't say nothing 'cos I feel like I'm not heard out. (ID08)

Similarly, some carers reported that their concerns had been 'not believed' (FC09) or 'dismissed as trivial and unimportant' (PC09). Having proposed their own ideas about medication, some carers reported being given a sense that it was not their place to do so:

The consultant was like 'you're talking rubbish' … it was like, 'what does she know?' (PC02)

I suggested a medication which had been mentioned previously and I had looked up the research on it. It's something that's very useful for people with high levels of anxiety and I thought it might be worth trying but umm … there was a small flicker and then, like, 'no, I don't think so, where did you hear about this?' sort of thing. (FC05)

Such experiences were reported to have contributed to family carers becoming burnt-out and resigning themselves to a subordinate position with respect to medication decisions. After what she described as a long and turbulent relationship with her relative's care team, one mother reluctantly stepped back from taking a more

active role in treatment decisions, stating 'we're [now] leaving it to them, I think that's the best way' (FC06).

Given their perception of being 'low ranked' in the hierarchy of stakeholders ('just a provider' (PC08) and 'not seen as a professional or intellectual resource' (PC11)), paid carers often felt the need to prove the credibility of their knowledge in order to be heard and have influence. Investing in the relationship with the psychiatrist was felt to make this easier ('because they know me, they know my information is really important' (PC05)), and paid carers sometimes sought legitimacy by presenting themselves as objective, collecting data and taking 'a paper trail … [of] evidence' (PC08) to appointments to support their views.

## DISCUSSION
### Principal findings
This qualitative study has enabled us to gain a deep understanding of the views and experiences of people with ID and their carers about psychotropic drug use and decision-making. Though highly topical given the prevalence of psychotropic prescribing in this group, the subject has been relatively little studied using qualitative approaches. The inclusion of multiple stakeholders adds an additional dimension to medication decision-making which we have been able to explore. Although preferences for involvement varied between individuals, most participants in our study valued having a place in decision-making. Experiences that were not aligned with expectation of involvement could lead to a range of emotional responses and prompt various efforts to gain position and influence.

People with ID reported having few opportunities to become involved in the psychotropic medication decision-making process. Only a minority described consciously ceding control to others, with most either unaware they were entitled to a role in deciding medication, or having been unsuccessful in involving themselves despite their efforts. Lack of knowledge about medication, a strong belief in medication as necessary and important, fear of the consequences of not taking medication (particularly admission to hospital), trust in the doctor as an expert and deference towards authority figures all underpinned a passive compliance and largely unquestioning stance towards medication. In this regard, our analysis supports the 'model of compliance' proposed by Crossley and Withers in their exploration of the experiences of people with ID prescribed antipsychotic medication,[28] and renews calls for greater efforts to inform and involve people with ID about their medication.

Family and paid carer groups, meanwhile, clearly expressed a desire to be involved in medication decision-making. This was related to a self-identity as the 'front-line people' and was intertwined with their often conflicted or uneasy attitude towards psychotropic medication. The carers strongly believed in the value of the contribution they could make to medication decisions, and considered their involvement essential to achieving the best outcome for the individual they supported. Positive experiences were described in terms compatible with collaborative and negotiated models of decision-making, although with the over-riding assumption that the psychiatrist would take final responsibility for prescribing decisions. While experiences of SDM undoubtedly did exist, these could not be taken for granted, and many study participants felt they had been denied a place in decision-making. Beneath this could be the devaluing of carer knowledge (based heavily on relational lived experience) in comparison to the technical knowledge and scientific expertise of the psychiatrist. This 'epistemic injustice'[35] prompted numerous attempts to rebalance the perceived power asymmetry in consultations as people tried to leverage influence or strengthen their voice. Although these could be successful to an extent, they required resources that were not available to all, added to the emotional toll of caring, and had caused some to lose faith in services.

### Clinical implications
The over-use of psychotropic medication for people with ID is now well evidenced and is the focus of national attention. Off-label prescribing, psychotropic polypharmacy and lengthy durations of medication treatment were all reported by the participants recruited for this study. The average duration of psychotropic use in our sample was 16 years, and the prevalence of antipsychotic use far outweighed the presence of severe mental illness. The STOMP programme in England, established to address these issues, has not yet achieved wholesale reductions in use of antipsychotic medication[36] but an assessment of medication optimisation must include more than a crude count of prescriptions. Improving medication outcomes for individuals requires a person-centred approach to prescribing that includes partnership between stakeholders and consideration of patients' values and goals on an equal footing to the expertise and opinion of mental health professionals. These elements are part of broader attempts to support patient autonomy, and are embodied in the SDM model.

The adoption of SDM in routine mental healthcare has been slow[37] and although psychiatrists explicitly endorse the model,[38] micro-analytic studies of routine psychiatric consultations show that its principles are infrequently applied.[39–41] Issues of insight, fluctuating mental capacity associated with episodes of acute and severe mental ill-health, power differentials between patient and professional, and the background threat of compulsory treatment, have all been identified as implementation barriers that are especially pertinent in psychiatric practise.[42] Arguably the challenges to SDM are compounded in people with ID[43 44] due to the fixed cognitive deficit, additional communication needs, and people's lack of experience and confidence in making choices about their healthcare or, indeed, more generally.[45 46]

The presence of multiple stakeholders adds an extra dimension to the SDM model, which has largely been developed with reference to dyadic doctor–patient

interactions and may not adequately account for complex decisions that are distributed within social networks.[42] Defining roles and responsibilities, and balancing the relative influence of different (and possibly conflicting) views adds to the challenges of achieving shared decisions in this group. Thus, if we are to achieve successful SDM, and in so doing, obtain its benefits, the model may need to be broadened.

A parallel concept of *supported* decision-making has been advanced for those with cognitive impairment,[47] and is similarly predicated on the principles of autonomy and self-determination. Supported decision-making formalises the place of a network of individuals, which may consist of family members, friends or other trusted people, who are able to help the person to formulate and express their preferences and thus exercise their autonomy. This may include assistance in gathering information, understanding their options and/or communicating their choice. Clearly, such tasks were often undertaken by carers interviewed in the present study and suggest that elements of the framework could be incorporated to an adapted model of SDM.

Increasing inclusion of people with ID and their paid and/or family carers in decisions (under whatever model this is branded) may represent a significant role change for all stakeholders. Clinicians, which our study indicates hold the majority of the decision-making power in these clinical encounters, will need to find ways of making conversations more accessible and collaborative as patient involvement becomes a legal as well as an ethical imperative.[48] People with ID must be made aware of their rights and appropriately supported in contributing to healthcare decisions to a level which they are comfortable with, if we are to avoid making unreasonable demands that risk alienating them from professionals. As we have reported, carers can play a pivotal role in contributing to this involvement, and this should be recognised and itself supported.

### Future work

Observing interactions within real-world consultations could lead to a more nuanced understanding of how medication discussions happen, and help to further develop theoretical models of healthcare decision-making in people with ID. Developing scalable interventions based on this understanding could improve opportunities for involvement of adults with ID and their carers. Several interventions have been developed and evaluated in people with mental health problems without ID.[49–53] Exploring the views of prescribers and other health professionals also is important and could uncover other factors that influence patient and carer involvement and which themselves could be a target for intervention. Finally, it will be necessary to demonstrate that incorporating SDM principles in routine care in this group is associated with improved patient-reported and objective outcomes.

### Strengths and limitations of this study

This study is unique in providing a multistakeholder analysis of accounts of the use of psychotropic medications in people with ID. It extends the existing qualitative literature in this field which has typically focused solely on antipsychotic drugs[28] or medication used for behaviour that challenges.[29 54 55] Synthesising the results of interviews with patients, family carers and paid carers allowed us to develop broad, over-arching themes, and helps us to understand the interactions and dynamics involved in the complex process of medication decision-making. Adaptations to the research method enabled us to gain meaningful insights into the experiences of people with ID, a group who are often excluded from research participation and may be considered inappropriate for in-depth qualitative investigation.[56] A relatively large sample size, with respondents purposively sampled from different locations and according to demographic and clinical characteristics, adds to the breadth of our findings.

The views of people with ID and their carers are difficult to obtain and seldom heard in the research literature. In prioritising their accounts, this research report does not include the views of general practitioners, pharmacists or psychiatrists. Participants were self-selecting and may have included only those with greater confidence. Their views are not necessarily representative of a wider group of people with ID and their carers. We only interviewed people (and carers of people) who were currently prescribed psychotropic medication and under the care of specialist psychiatry teams, thereby excluding those who may have previously taken medication, been managed solely in primary care or who have chosen not to take medication for mental health problems. People in any of these groups may possess different and equally-valid perspectives on psychotropic medication and its prescribing.

### CONCLUSION

Achieving optimal use of psychotropic medication is a health service priority and can only occur when working in partnership with people with ID and their carers. Frameworks such as SDM which are based on the principles of personalisation and collaboration offer a possible means of ensuring that stakeholders are represented in important decisions. Our study suggests that successful collaborative decisions regarding medication are achievable but are not always experienced. Further research to understand how medication decisions are made from the perspective of prescribers and how other stakeholders can be meaningfully and productively included is necessary to inform the development of interventions that help ensure people with ID and their carers have a true voice in medication discussions and decisions.

**Acknowledgements** The authors would like to thank those who agreed to take part in the study and the organisations and individuals that assisted with

recruitment, and Restu Handoyo, who contributed to the analysis. We also thank members of the service user consultation group, Jackie McMorrow and Jill Huntesmith.

**Contributors** RS, AH, AS and NM designed the study. RS recruited to the study and carried out the interviews. RS, AH, AS and NM undertook the analysis. RS and NM drafted the manuscript with input from AH and AS. All authors approved the final version.

**Funding** This study was funded by a Doctoral Research Fellowship awarded to RS from the National Institute for Health Research (NIHR) (Ref: DRF-2016-09-140).

**Disclaimer** The views expressed in this article and those of the authors and not necessarily those of the NIHR or the Department of Health and Social Care. The funder had no role in study design, analysis, decision to publish, or preparation of the manuscript.

**Competing interests** None declared.

**Patient consent for publication** Not required.

**Ethics approval** The study was approved by the London-Surrey NHS Research Ethics Committee (reference 17/LO/1365). Local Research and Development approvals were obtained prior to any research activities being undertaken.

**Provenance and peer review** Not commissioned; externally peer reviewed.

**Data availability statement** No data are available.

**ORCID iDs**
Rory Sheehan http://orcid.org/0000-0002-4164-9661
Nicola Morant http://orcid.org/0000-0003-4022-8133

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
