## [Reviewer comments · BMJ Open]

ARTICLE DETAILS

TITLE (PROVISIONAL)	Experiences of psychotropic medication use and decision-making for adults with intellectual disability: a multi-stakeholder qualitative study in the UK
AUTHORS	Sheehan, Rory; HASSIOTIS, ANGELA; Strydom, André; Morant, Nicola

VERSION 1 – REVIEW

REVIEWER	Dr Richard Keers Clinical Lecturer in Pharmacy, The University of Manchester, UK. Honorary Research Pharmacist, Greater Manchester Mental Health NHS Foundation Trust, UK.
REVIEW RETURNED	26-Jul-2019

GENERAL COMMENTS	Many thanks for submitting this interesting and well written manuscript for publication. It addresses an important research question relevant to current clinical practice, and gives a platform to the voices of patients and carers which are vital in efforts to improve health care outcomes. At this point, I would like to say that my review of this manuscript was conducted using my background as a mental health pharmacist and qualitative researcher. I have awareness of the fields of intellectual/learning difficulties and shared decision making, but do not possess expertise in these areas. I thought that the study was well conducted and reported, and my comments mainly relate to the need for additional detail and clarification in areas. I have therefore provided these by section below. However, before this point I would like to raise a few important issues that require specific attention from the authors. First, I could find no mention of whether research ethics was required for this project, which it most certainly would have been. Can the authors please add details of ethical and R&D approvals (with approval numbers) to the manuscript, and mention in the abstract. Second, in the methods the authors mention that they only interviewed participants who had mental capacity and sufficient communication ability, but that they didn't assess this themselves. The authors should please make clear in the manuscript how exactly they ensured that their participants met these criteria. This is very important as the authors themselves highlight fluctuating mental capacity as a consideration for SDM on page 31. Third, the findings indicated that carers (and particularly family carers) advocated and represented those they cared for. There were some indications also in the results that carers took control of
---

	care for patients and acted on their behalf. The authors indicate some of the possible reasons for this, but can they please use an alternative word to 'possible' that is grounded in the data - if these reasons are not clear please state so and discuss later in the paper. This feels like another dimension of shared decision making could exist between the patient and their carer in relation to medication decisions, and I would like the authors to please consider including this in their results/discussion if the data supports it. For example, did family carers make shared decisions with the patient on these issues or not? Did patients feel like they were excluded from conversations between their carer and health care professional? Fourth, in Table 1 there is some data presented with n=1 participants. I would strongly recommend combining groups together to avoid n=1 as otherwise there is an increased risk of individual participants being identifiable from the data. Fifth, I can see that the authors have referred specifically to SDM in the introduction and discussion as a means to help give context to their findings. what is unclear is whether this influenced the preparation of their topic guide (if so please state) and subsequent thematic analysis. If this is not the case, I am wondering why SDM has become a key component of the paper despite not being part of the methodology - there may be other approaches to enhance involvement in decision making and I think that the authors should present SDM in context of any other approaches in these sections of the report. The findings do not necessary indicate that SDM must be the approach to use in future, but instead indicate that an approach like SDM may be explored (along with other relevant options). TITLE/ABSTRACT  - On balance, the results section is dedicated as much (if not more) to more general experiences of psychotropic medications as decision making. Can the authors please amend the title and abstract to reflect this focus Please can the authors add/change the following the details to the abstract to improve transparency:  - relevant approvals to be added - please consider the use of terminology for consistency in the paper as a whole - patients or people, doctor or prescriber? - please refer to 'adult' patients in the objectives and participant sections (and if necessary elsewhere in the paper) - please elaborate on whether 1+ authors independently confirmed the coding framework - please provide a little more detail of the recruitment approach here, such as advertisement etc. - Results, please change to: People with ID reported being highly compliant... - Most of the results section focuses on carers rather than those with ID. Can the authors please consider adding a little more patient perspective - there were some positive experiences of care/shared decision making reported in the results which should please be briefly mentioned in the results/conclusion - Conclusion: please make reference to the main findings of the project as the first sentence. Please change: Shared decision making is a model that could offer...
--	--

STRENGTHS/LIMITATIONS

- Please amend to: power dynamics underpinning decision making in the context of psychotropic medication use.
- Please remove 'and' between 'work and extends'

INTRODUCTION

- Page 6, please amend paper to 'owned jointly by health professional and patient.'
- Page 6, Please present evidence of the impact of SDM on outcomes in the paper, e.g. adherence, knowledge/perceptions/attitudes, appropriate prescribing. This is essential to justify its inclusion as a means by which to frame the qualitative findings of the study
- Page 6, final paragraph, 'little exploration'. Can the authors please qualify what this means, if there are papers please cite them or if not say there are none. For example could the authors reference studies 27,28 and describe their relationship to the current aim/work, and/or perhaps discuss any evaluations from the staff perspective.

I came across the following text which might be helpful:

<https://www.researchgate.net/publication/319188533>

Best_Practice_Health_Care_Decision_Making_By_With_and_For_Adults_with_Learning_Disabilities

- within the introduction there is discussion of SDM without much mention of supported decision making, which is a related field and has a well developed evidence base. Can the authors please briefly distinguish between the two in the paper and highlight how this study focuses on one area rather than the other (or both?)

METHODS

- Page 7, please amend to; 'who were currently prescribed psychotropic...'
- Page 7, please clarify in the paper how psychotropic medication was defined
- Page 7, please elaborate on the settings mentioned 'variety of settings' in the text.
- Page 7, please indicate in the text who offered the leaflet and how/where this took place. Please also indicate how many regions of England this project took place in, and name them in general terms (e.g. North, South). The authors mention presentations in the third sector, were any other methods used (e.g. leaflets?) if so please report.
- Page 7. Please clarify in the text how potential participants indicated interest in the project and how they were then contacted to confirm participation. E.g. completing consent to contact form?
- Page 7. Please clarify in the text whether patients were only sought if they were under the care of a psychiatrist. This would help explain the lack of reference to GP consultations in the results (who can of course newly prescribe antidepressant treatment in the UK).
- Page 8. First two lines (characteristics). Please clarify the characteristics sought for patients and those for carers, as these may differ (e.g. psychiatric morbidity does not apply to carers, and is more related to the people they care for).
- Page 8. Please clarify in the text what this means: 'Baseline demographics and descriptive data were collected by participant report.'
- Page 8. Please confirm in the text whether interviews could be face-to-face or telephone, and whether participants could bring people along to their meetings (and also whether group interviews with patients and carers together was possible).

- Page 8. Please amend to: first author, who was a psychiatrist...
- Page 8. Please provide in the text a brief written summary of the main question domains in the interview schedule. Please also provide the interview schedule as a supplementary file as this will help the reader to understand the purpose of the study and adequately frame the results.
- Page 8. Please clarify in the text whether field notes were taken with the permission of the interviewee.
- Page 8. Please clarify in the text where interviews could take place.
- Page 9. Please clarify in the text exactly how many transcripts were independently coded, and by whom. Was this person a member of the research, and if not why not?
- Page 9. Please amend to: 'The development of the recruitment strategy, participant materials,...'

RESULTS

Page 13: please amend to 'that were described amongst the group...'

Page 14. With regards to the process of carers translating information between doctor and patient, all the quotes used were from paid carers. Can the authors please include the family carer perspective on this issue, and highlight if it diverges

- Page 15. can the authors please add why paid carers were hesitant in offering their own opinions on medication?
- Page 23. 'The most fundamental...' Was this the opinion of the authors or the participants? Please qualify in the text and move to discussion if it is the authors opinion.
- Page 23. Please amend to: 'Both paid and family carers reporting being deprived of information...'
- Page 24. Please amend to: 'This knowledge was reported by participants to improve their confidence and go...' . Also, 'Secondly, respondents...'
- Page 25. Please use an alternative word to 'shun', e.g. avoid
- Page 26. Can the authors please elaborate on the situations where influence was achieved successfully? This will help balance this section of the report.

DISCUSSION AND CONCLUSION

- For the first paragraph of your discussion, it would be helpful for the authors to first describe using summary format the overall aim of their work, if that aim was met and to emphasize the key findings and their place in the context of the body of research in the field (e.g. first paper to...).

- Page 29. Please amend in text to: People with ID reporting having few opportunities...' 'Doctor as an expert...'

- It is important to balance in the discussion any positive experiences and what can be learnt from these against negative experiences and the areas for improvement - whilst the findings do clearly support need for improvement, the overall impression is quite negative when this was not the complete story presented in the results

- Page 31. Please amend to 'Shifting the paradigm to SDM may therefore represent...'

Please amend to: 'Clinicians, which our study indicates hold the majority...'

- Page 33. In future work, the authors should discuss the importance of obtaining the health care professional perspective on this topic with reference to any existing literature on the topic, why

	this work needs to be carried out and what this will potentially add to the current study and field as a whole (e.g. training needs). - Page 33 and Conclusion. The authors should please add that future work should include the need to develop model(s) of what SDM in the context of those with ID and mental health needs might look like, before implementation and evaluation. - Page 33. Please remove 'ethnographic work' as this approach is not necessary to answer the research question being posed, and is a very specific type of research that is difficult to conduct. I suspect that non-participant observation of the use of audio-video recording may suffice
--	---

REVIEWER	Philip McCallion Temple University, USA
REVIEW RETURNED	25-Aug-2019

GENERAL COMMENTS	in the midst of implementation of STOMP this is a timely paper capturing the "state of play" of involvement of people with ID and their carers  1. A rationale for not including psychiatrists/GPs should be included since there was little reference to literature on their thoughts on roles 2. SDM would benefit from inclusion in ideas of person-centered planning - might explain the disconnect where carers in particular and an increasing number of people with ID feel they should be involved not simply consulted about care decisions 3. Would have been useful to ask paid carers to identify characteristics of people cared for as their responses were about specific situations 4. Just an observation - paid carers description of their role as facilitative was "the right answer" but much of the literature speaks of family-like relationships for paid carers where care is extended. 5. More information on the interview protocol and approach would help potential for replication 6. Ethics not really addressed. 7. Unclear in discussion what are the practice implications for psychiatry 8. Some stray words in several places should be addressed 9. some more up to date references on SDM with people with ID
---

VERSION 1 – AUTHOR RESPONSE

Reviewer: 1

I thought that the study was well conducted and reported, and my comments mainly relate to the need for additional detail and clarification in areas. I have therefore provided these by section below. However, before this point I would like to raise a few important issues that require specific attention from the authors.

We thank the reviewer for their close attention to the paper and their helpful comments. We have responded to each in turn.

First, I could find no mention of whether research ethics was required for this project, which it most certainly would have been. Can the authors please add details of ethical and R&D approvals (with approval numbers) to the manuscript, and mention in the abstract.

Thank you for highlighting this. The research was approved by the London-Surrey NHS Research Ethics Committee (REC reference 17/LO/1365) and all relevant local R&D approvals were obtained

prior to any research activities being conducted. We now include this on page 9 of the updated manuscript.

Second, in the methods the authors mention that they only interviewed participants who had mental capacity and sufficient communication ability, but that they didn't assess this themselves. The authors should please make clear in the manuscript how exactly they ensured that their participants met these criteria. This is very important as the authors themselves highlight fluctuating mental capacity as a consideration for SDM on page 31.

Eligibility to take part in this study was first assessed at the point of identification of potential participants by researchers or clinicians who made the initial approach and informed people of the research. This assessment continued (e.g. in liaising with potential participants and / or carers) prior to interviews being held.

Capacity to consent to taking part in the research was assessed immediately before the interview, as part of the consent procedure and in accordance with the Mental Capacity Act (2005). The researcher who conducted the capacity assessment and consent procedure is also a clinician working in the field of ID and has experience and professional training in assessing capacity in people with cognitive impairment. It was made clear to participants that their contribution was voluntary, that they could decline without prejudice, and they may terminate the interview at any time. We have clarified this procedure in the methods, with additional information on recruitment methods (page 8).

Third, the findings indicated that carers (and particularly family carers) advocated and represented those they cared for. There were some indications also in the results that carers took control of care for patients and acted on their behalf. The authors indicate some of the possible reasons for this, but can they please use an alternative word to 'possible' that is grounded in the data - if these reasons are not clear please state so and discuss later in the paper.

Thank you for highlighting this, which we assume relates to the sentence starting "possibly owing to differences in the degree of ID of those they cared for..." We have now removed the beginning of this sentence as we agree this is an interpretation, rather than being grounded in the data. The reasons for paid and family carers assuming different approaches or perspectives were not discussed at length and we have now not attempted to speculate on this.

This feels like another dimension of shared decision making could exist between the patient and their carer in relation to medication decisions, and I would like the authors to please consider including this in their results/discussion if the data supports it. For example, did family carers make shared decisions with the patient on these issues or not? Did patients feel like they were excluded from conversations between their carer and health care professional?

There were carer accounts of providing support to the person with ID to enable them to understand their medication, increase their autonomy, and be more involved in the process of decision-making and we now included a quotation to support this (page 21). There was no evidence in the data that patients felt excluded from conversations between their carer and healthcare professional.

We have also added to the final paragraph of the introduction to give earlier prominence to the stakeholders in medication decisions for people with ID (page 7).

Fourth, in Table 1 there is some data presented with n=1 participants. I would strongly recommend combining groups together to avoid n=1 as otherwise there is an increased risk of individual participants being identifiable from the data.

We have changed the table with respect to relation of the family carer. We did not feel it was appropriate to further collapse the ethnicity variable (page 13).

Fifth, I can see that the authors have referred specifically to SDM in the introduction and discussion as a means to help give context to their findings. What is unclear is whether this influenced the preparation of their topic guide (if so please state) and subsequent thematic analysis. If this is not the case, I am wondering why SDM has become a key component of the paper despite not being part of the methodology - there may be other approaches to enhance involvement in decision making and I think that the authors should present SDM in context of any other approaches in these sections of the report. The findings do not necessarily indicate that SDM must be the approach to use in future, but instead indicate that an approach like SDM may be explored (along with other relevant options).

We thank the reviewer for these comments. We recognise that SDM is not the only model aiming to enhance patient involvement in treatment decisions. We have revised the introduction to introduce other related ideals and concepts, all of which are aimed at supporting patient autonomy and reducing paternalism in healthcare (page 6).

We have also revised the discussion and conclusions. We now present SDM as a potential framework through which increased patient and carer involvement may be achieved, but consider this more broadly and include a discussion of other models, for example, supported decision making (as the reviewer suggests in a later point).

As with any qualitative work, we recognise that our theoretical, professional and personal positions may shape how we design and conduct research, and aim to be reflexive about this. We are interested in the principles related to SDM and clinical improvements it may encourage, but are not wedded to any particular model within which these are framed.

TITLE/ABSTRACT

- On balance, the results section is dedicated as much (if not more) to more general experiences of psychotropic medications as decision making. Can the authors please amend the title and abstract to reflect this focus.

We have amended the title to house style, also suggested by the editor.

Please can the authors add/change the following the details to the abstract to improve transparency:

- relevant approvals to be added

The ethics approvals have been added to the manuscript (p XX) but word count limitations preclude adding this information to the abstract.

- please consider the use of terminology for consistency in the paper as a whole - patients or people, doctor or prescriber?

We refer to "people with ID" in most cases and "patients" where clearly in this role.

- please refer to 'adult' patients in the objectives and participant sections (and if necessary elsewhere in the paper)

We have made the suggested change.

- please elaborate on whether 1+ authors independently confirmed the coding framework

We have added more detail about the analytic process in the main body of the manuscript, and describe these changes more fully in our response to Reviewer 1's comment about how data analysis is described in the methods. Abstract word count limitations preclude more detail being given here.

- please provide a little more detail of the recruitment approach here, such as advertisement etc.

The study used two methods of recruitment.

- (1) Potential participants were approached by their usual clinician in the course of routine clinical contacts, and provided with a leaflet about the research. If they were interested in hearing more, they consented to their details being passed to the research team, who contacted them at a later date.
- (2) Potential participants were also reached through presentations to relevant community groups and leaflets about the research were available. Those who were interested could give their details to the research team at these presentations or contact the research team at a later date.

We have added to the methods section but are unable to add this level of detail to the abstract.

- Results, please change to: People with ID reported being highly compliant...

We have made the suggested change.

- Most of the results section focuses on carers rather than those with ID. Can the authors please consider adding a little more patient perspective

We have re-worked the results section of the abstract to include more of the perspective of those with ID.

- there were some positive experiences of care/shared decision making reported in the results which should please be briefly mentioned in the results/conclusion

We have added to the results section to make this point.

- Conclusion: please make reference to the main findings of the project as the first sentence. Please change: Shared decision making is a model that could offer...

We have referenced the overall findings of the project in the first sentence, while keeping the conclusions broad and being mindful of the space limitations of the abstract.

STRENGTHS/LIMITATIONS

- Please amend to: power dynamics underpinning decision making in the context of psychotropic medication use.

We have made the suggested change.

- Please remove 'and' between 'work and extends'

Thank you for highlighting this typo which we have now amended.

INTRODUCTION

- Page 6, please amend paper to 'owned jointly by health professional and patient.'

We have made the suggested change.

- Page 6, Please present evidence of the impact of SDM on outcomes in the paper, e.g. adherence, knowledge/perceptions/attitudes, appropriate prescribing. This is essential to justify its inclusion as a means by which to frame the qualitative findings of the study

We have added context that SDM is associated with certain improved outcomes (page 6).

- Page 6, final paragraph, 'little exploration'. Can the authors please qualify what this means, if there are papers please cite them or if not say there are none. For example could the authors reference studies 27,28 and describe their relationship to the current aim/work, and/or perhaps discuss any evaluations from the staff perspective.

I came across the following text which might be helpful:

https://eur01.safelinks.protection.outlook.com/?url=https%3A%2F%2Fwww.researchgate.net%2Fpublication%2F319188533_Best_Practice_Health_Care_Decision_Making_By_With_and_For_Adults_with_Learning_Disabilities&data=02%7C01%7C%7C24df639d8e364c2a916b08d72ae21f94%7C1fa88fea9984c5b93c9210a11d9a5c2%7C0%7C0%7C637025022986774995&data=1FShVeQmL2AmLvbEnxEsbKQXJ8Bfve4nOaOZgEXmYQ0%3D&reserved=0

We have moved some information and references from the discussion to the introduction in response to this point and now give extra detail of the studies conducted in this area. There is evidence that people with ID and their carers are not centred in healthcare decisions, in general, but the literature specifically related to involvement in psychotropic medication decisions is less developed. We thank the reviewer for the suggested reference which is a report of the article by Fovargue et al, 2000, which we originally cited but have now replaced.

- within the introduction there is discussion of SDM without much mention of supported decision making, which is a related field and has a well developed evidence base. Can the authors please briefly distinguish between the two in the paper and highlight how this study focuses on one area rather than the other (or both?)

Thank you for this suggestion. We have added to the introduction to explain the overlaps in concepts related to increasing patient involvement that appear in policy rhetoric and good practice guidelines (page 6). We now include a paragraph on supported decision-making to the discussion, and consider how this may be similar to, or complement, shared decision making (page 35-36).

METHODS

- Page 7, please amend to; 'who were currently prescribed psychotropic...'

We have made this suggested change.

- Page 7, please clarify in the paper how psychotropic medication was defined

Psychotropic medication was defined as any drug listed in the British National Formulary as being used for mental health disorders (Joint Formulary Committee (2019). BNF 77: March 2019. London: Pharmaceutical Press). We have added this detail to the text (page 7).

- Page 7, please elaborate on the settings mentioned 'variety of settings' in the text.

Paid carers may have been employed in a variety of settings including residential homes, supported living projects, or as peripatetic community support workers. We have added this to the text (page 7)

- Page 7, please indicate in the text who offered the leaflet and how/where this took place. Please also indicate how many regions of England this project took place in, and name them in general terms (e.g. North, South). The authors mention presentations in the third sector, were any other methods used (e.g. leaflets?) if so please report.

The study was conducted in the south-east of England (page 8). Leaflets with details of the research team were available at presentations.

- Page 7. Please clarify in the text how potential participants indicated interest in the project and how they were then contacted to confirm participation. E.g. completing consent to contact form?

Clinicians passed contact details of potential participants to the research team, with their consent. People who were made aware of the research at presentations could give their details to the research team directly (page 8). Having received the leaflet, potential participants could also contact the research team.

- Page 7. Please clarify in the text whether patients were only sought if they were under the care of a psychiatrist. This would help explain the lack of reference to GP consultations in the results (who can of course newly prescribe antidepressant treatment in the UK).

Yes, this was an inclusion criterion which we have now clarified in the text (page 7). This is also in the limitations section of the discussion as we recognise this criterion will influence the data collected (page 38).

- Page 8. First two lines (characteristics). Please clarify the characteristics sought for patients and those for carers, as these may differ (e.g. psychiatric morbidity does not apply to carers, and is more related to the people they care for).

Purposive sampling was used to select participants with a range of characteristics that may be related to medication views and experiences. For people with ID this included, age, gender, ethnic group, indication for psychotropic medication and medication class; for family carers, age, gender, ethnic group, degree of ID in their relative, indication for and class of medication; and for paid carers, age, gender, ethnic group, duration working with people with ID, and seniority. We have added this to the text (page 8-9).

- Page 8. Please clarify in the text what this means: 'Baseline demographics and descriptive data were collected by participant report.'

Participants reported these characteristics and we did not cross-check these against other sources of information e.g. clinical records. We have added to the text to make this clear (page 9).

- Page 8. Please confirm in the text whether interviews could be face-to-face or telephone, and whether participants could bring people along to their meetings (and also whether group interviews with patients and carers together was possible).

All interviews were conducted face-to-face (now page 9). People could bring others to their interviews, and 7 people with ID did this (page 10 and page 12). Group interviews between people with ID and carers, were not undertaken.

- Page 8. Please amend to: first author, who was a psychiatrist...

We have made this change.

- Page 8. Please provide in the text a brief written summary of the main question domains in the interview schedule. Please also provide the interview schedule as a supplementary file as this will help the reader to understand the purpose of the study and adequately frame the results.

We have added to the text and now include the topic guides as supplementary material.

- Page 8. Please clarify in the text whether field notes were taken with the permission of the interviewee.

These were reflective field notes to capture the interviewer's impressions, meanings, and ideas arising from the data were made immediately after interviews.

- Page 8. Please clarify in the text where interviews could take place.

We have added to the methods that interviews were held at a time and place preferred by participants (page 10). More detail is included in the results – interviews were conducted at people's home, place of work, university, or community centres (page 12).

- Page 9. Please clarify in the text exactly how many transcripts were independently coded, and by whom. Was this person a member of the research, and if not why not?

We have added details of this process to the description of the data analysis (page 10). Analysis involved close collaboration with the research team and one other (a researcher in a related field). This enhanced analysis by drawing on the perspectives and interpretations of people both 'close to' and more distant from the research study and its focus.

- Page 9. Please amend to: 'The development of the recruitment strategy, participant materials,...'

We have made this change.

RESULTS

Page 13: please amend to 'that were described amongst the group...'

We have made this change.

Page 14. With regards to the process of carers translating information between doctor and patient, all the quotes used were from paid carers. Can the authors please include the family carer perspective on this issue, and highlight if it diverges

Family carers also acted to interpret and translate information. We have added a quote from a family carer to illustrate this.

- Page 15. can the authors please add why paid carers were hesitant in offering their own opinions on medication?

That paid carers were less forthcoming in offering their views about psychotropic medication was an interesting finding. Unfortunately the reasons for this are not clear in the data, and as such, we feel it is not appropriate to comment further on this in the results.

- Page 23. 'The most fundamental...' Was this the opinion of the authors or the participants? Please quality in the text and move to discussion if it is the authors opinion.

Thank you for pointing this out. We have changed 'fundamental' to 'pre-requisite' (page 26).

- Page 23. Please amend to: 'Both paid and family carers reporting being deprived of information...'

We have made this change.

- Page 24. Please amend to: 'This knowledge was reported by participants to improve their confidence and go...' . Also, 'Secondly, respondents...'

We have made this change.

- Page 25. Please use an alternative word to 'shun', e.g. avoid

We have made this change.

- Page 26. Can the authors please elaborate on the situations where influence was achieved successfully? This will help balance this section of the report.

We have re-worked the first part of the decisional processes theme (starting page 21). As well as changing the sub-title from 'unequal power dynamics' to 'power dynamics', we have explained in more detail the level of involvement that people valued, and given more examples of experiences where the decision-making approach was aligned with these preferences. These experiences tended to involve having time to talk and a good working relationship with the psychiatrist.

DISCUSSION AND CONCLUSION

- For the first paragraph of your discussion, it would be helpful for the authors to first describe using summary format the overall aim of their work, if that aim was met and to emphasize the key findings and their place in the context of the body of research in the field (e.g. first paper to...).

We have added a paragraph to this effect (page 32).

- Page 29. Please amend in text to: People with ID reporting having few opportunities...' 'Doctor as an expert...'

We have made this change.

- It is important to balance in the discussion any positive experiences and what can be learnt from these against negative experiences and the areas for improvement - whilst the findings do clearly support need for improvement, the overall impression is quite negative when this was not the complete story presented in the results

We have reviewed the paper thoroughly and made appropriate changes in the results and discussion to add balance throughout. We hope the reviewer will also agree that we are now less definitive in our discussion/conclusions, both in the interpretation of the results and in proposing SDM as a model for practice change.

- Page 31. Please amend to 'Shifting the paradigm to SDM may therefore represent...'
Please amend to: 'Clinicians, which our study indicates hold the majority...'

We have made these changes to these sentences.

- Page 33. In future work, the authors should discuss the importance of obtaining the health care professional perspective on this topic with reference to any existing literature on the topic, why this work needs to be carried out and what this will potentially add to the current study and field as a whole (e.g. training needs).

We agree with the reviewer in the need to hear the perspectives from additional stakeholder groups. We have added this to the paragraph describing future work (page 37) and also mention this in the limitations of the paper (page 38).

- Page 33 and Conclusion. The authors should please add that future work should include the need to develop model(s) of what SDM in the context of those with ID and mental health needs might look like, before implementation and evaluation.

We accept that the literature around SDM or other models of improving the involvement of people with ID is at an early stage and have moderated the conclusions of this study accordingly. For example, we no longer speak of embedding and implementing SDM in this context and focus more on understanding decision-making and developing means of improving involvement of stakeholders.

- Page 33. Please remove 'ethnographic work' as this approach is not necessary to answer the research question being posed, and is a very specific type of research that is difficult to conduct. I suspect that non-participant observation of the use of audio-video recording may suffice

We have removed the suggestion to conduct ethnographic work.

Reviewer: 2

In the midst of implementation of STOMP this is a timely paper capturing the "state of play" of involvement of people with ID and their carers

1. A rationale for not including psychiatrists/GPs should be included since there was little reference to literature on their thoughts on roles

There are many stakeholders in this sphere and we agree that it is important to hear from all as many as possible. This project forms part of a larger programme of work in which we also obtained the views of psychiatrists. We will be reporting these data elsewhere. We acknowledge this as a limitation of the present analysis (page 38).

2. SDM would benefit from inclusion in ideas of person-centered planning - might explain the disconnect where carers in particular and an increasing number of people with ID feel they should be involved not simply consulted about care decisions

We thank the reviewer for this point and a similar comment was made by reviewer 1. We have added to the introduction to mention the overlapping terms that are in use which all have common underlying principles, including person-centred care (page 6). We have reduced the emphasis placed

on SDM accordingly, and now discuss the results in broader terms and in the context of increasing patient inclusion and autonomy.

3. Would have been useful to ask paid carers to identify characteristics of people cared for as their responses were about specific situations

We were not able to collect this information as each paid carer reported experiences and attitudes formed from supporting many different people and an explanation of this is now included (page 10).

4. Just an observation - paid carers description of their role as facilitative was "the right answer" but much of the literature speaks of family-like relationships for paid carers where care is extended.

The nature of the relationship between the paid carer and the person with ID they supported is interesting. We did find that paid carers were, in general, more 'reserved' in offering their personal opinion about medication, which could suggest that they valued a more 'objective' stance. There was not any suggestion in the data that relationships between the paid carer and the person with ID they supported changed or became more 'family-like' over time.

5. More information on the interview protocol and approach would help potential for replication

We now include additional detail in the manuscript (page 10) and the topic guides as supplementary material.

6. Ethics not really addressed.

Thank you for highlighting this and apologies for the omission. We have now included the ethical approvals obtained in the manuscript text (page 9).

7. Unclear in discussion what are the practice implications for psychiatry

We have added to the discussion of challenges of implementing shared decision-making in this context and have discussed the potential development of the model with aspects of supported decision-making. As this is exploratory work, we are unable to provide definitive recommendations to inform immediate practice changes but the work has value in adding patient and carer voice, which is rarely heard.

8. Some stray words in several places should be addressed

We have been through the manuscript prior to re-submission to remove these, as well as making several changes to improve readability and comprehension.

9. Some more up to date references on SDM with people with ID

There is very little academic literature about the application of shared decision-making in people with ID. We have added to the breadth of literature cited in the introduction about healthcare decision-making in people with ID and carers (page 6-7).

VERSION 2 – REVIEW

REVIEWER	Richard Keers The University of Manchester, United Kingdom I am a Council Member for the UK charity the College of Mental Health Pharmacy and assist with organising their annual conference which is sponsored by the pharmaceutical industry. I have received payment for the development of learning materials in the area of mental capacity and covert administration that are distributed nationally by the Centre for Pharmacy Postgraduate Education (CPPE), in the UK.
-----------------	---

	I have received payment for learning materials I have developed and presented for primary care health professionals on the topic of schizophrenia for MORPh Consultancy Ltd, which are sponsored by the pharmaceutical industry.
REVIEW RETURNED	24-Oct-2019

GENERAL COMMENTS	The authors have carefully considered and thoroughly addressed my comments and the manuscript is now much improved. There are only very minor corrections now to make. My amendments relate to the version of the manuscript with tracked changes highlighted.  - Page 8 line 54: please change 'people with ID are not' to 'evidence indicates people with ID may not be placed...' as cannot definitively say that they are or not based on one study - Page 12 line 45: 'interview topics included...' - Page 14 line 57: 'peoples homes' please change - Page 15: the authors should justify why they have ethnic group n=1 listed for some participants and why they were not collapsed down as this could lead to potentially identifying participants. I recommend these should be collapsed down to help protect their identity - Page 18, line 36-37: please reword as parts not clear, e.g.: 'describing of adverse effects' - Page 20, line 50/51: 'render them almost incapable' is strong language and appears to be the researchers own opinion. Please change this or provide as a quote if a participant said it - Page 27, line 17/18: please remove 'obviously' as this is the researchers opinion. Please check the manuscript carefully for other examples of researcher bias entering the results as there are two examples I have found so far. - Page 30, line 40/41: please remove 'not' - Page 31, lines 13-16: please reword as difficult sentence to process - Page 35, lines 35/36: 'Some carers were' implies this definitely happened, but actually it is 'carers reported' . Please check results section and be careful not to assume things did or did happen, it is only what participants report. See also page 36, lines 12/13 - change 'could lead' to something that the participants reported - Page 37, first paragraph: Could the authors use this paragraph to highlight the main headline on what is new/added to the literature, that links to their introduction where they highlighted what gaps this project would address? Please also reword some parts to improve clarity, 'and made more so' and 'given the prevalence' (high prevalence? more than general population?). Please also review semi colon use in lines 36/37 as the sentences could be better structured. - Page 38, lines 26/27 - this does not make much sense that they want to be involved, are 'front line people' (remove use of one set of speech marks at front) but were ambivalent about medication? Was this something strongly presented in the results? Please review to confirm and consider rewording if required - Page 40, what does apropos dyadic mean? I would use alternative language - Page 42, lines 40/41 - please rephrase '...and which themselves be a target...' as it could be clearer - Page 44, lines 40/41: 'brought into this is necessary' please reword as lacking clarity , e.g. change to 'included' or 'integrated'? - Page 44, lines 45/46: please change to 'help ensure'
--

REVIEWER	Philip McCallion Temple University USA
REVIEW RETURNED	24-Oct-2019
GENERAL COMMENTS	I thank the authors for responding to the questions/comments previously raised

VERSION 2 – AUTHOR RESPONSE

Reviewer: 1

The authors have carefully considered and thoroughly addressed my comments and the manuscript is now much improved. There are only very minor corrections now to make. My amendments relate to the version of the manuscript with tracked changes highlighted.

We wish to thank the reviewer for their detailed comments. We have addressed each of these in turn, as highlighted in the manuscript marked copy (page references below relate to the marked version).

- Page 8 line 54: please change 'people with ID are not' to 'evidence indicates people with ID may not be placed...' as cannot definitively say that they are or not based on one study

We have made this change (page 6, line 51).

- Page 12 line 45: 'interview topics included...'

We have made this change (page 10, line 16).

- Page 14 line 57: 'peoples homes' please change

We have made this change (page 12, line 14).

- Page 15: the authors should justify why they have ethnic group n=1 listed for some participants and why they were not collapsed down as this could lead to potentially identifying participants. I recommend these should be collapsed down to help protect their identity

We categorised ethnicity in standard categories as we had aimed to avoid a binary classification with one ethnic group as the reference category. We feel that the risk of de-anonymising participants using these data is low. However, we have now changed the manuscript (page 12, lines 50-56), as the reviewer suggests (and added a footnote to the table), but would also like to invite the editor to give their opinion on this point.

- Page 18, line 36-37: please reword as parts not clear, e.g.: 'describing of adverse effects'

We have changed the sentence to improve clarity (page 15, line 39).

- Page 20, line 50/51: 'render them almost incapable' is strong language and appears to be the researchers own opinion. Please change this or provide as a quote if a participant said it

We have removed this phrase (page 17, line 51).

- Page 27, line 17/18: please remove 'obviously' as this is the researchers opinion. Please check the manuscript carefully for other examples of researcher bias entering the results as there are two examples I have found so far.

We have removed this word (page 23, line 47).

- Page 30, line 40/41: please remove 'not'

We have made this change (page 26, line 29).

- Page 31, lines 13-16: please reword as difficult sentence to process

We have changed this sentence (page 26, lines 56-57).

- Page 35, lines 35/36: 'Some carers were' implies this definitely happened, but actually it is 'carers reported'. Please check results section and be careful not to assume things did or did happen, it is only what participants report. See also page 36, lines 12/13 - change 'could lead' to something that the participants reported

Thank you for these comments which have been addressed (e.g. page 28, line 6; page 31, line 11) .

- Page 37, first paragraph: Could the authors use this paragraph to highlight the main headline on what is new/added to the literature, that links to their introduction where they highlighted what gaps this project would address? Please also reword some parts to improve clarity, 'and made more so' and 'given the prevalence' (high prevalence? more than general population?). Please also review semi colon use in lines 36/37 as the sentences could be better structured.

We have re-written the first paragraph of the discussion, as requested. It now ties-in with the stated aims and highlights the novelty of the work (pages 32-33).

- Page 38, lines 26/27 - this does not make much sense that they want to be involved, are 'front line people' (remove use of one set of speech marks at front) but were ambivalent about medication? Was this something strongly presented in the results? Please review to confirm and consider rewording if required

We have changed this sentence in view of this comment (page 33, lines 45-50).

- Page 40, what does apropos dyadic mean? I would use alternative language

We have made this change (page 35, line 45).

- Page 42, lines 40/41 - please rephrase '...and which themselves be a target...' as it could be clearer

We have made this change (page 37, line 27).

- Page 44, lines 40/41: 'brought into this is necessary' please reword as lacking clarity, e.g. change to 'included' or 'integrated'?

We have made this change (page 39, line 13).

- Page 44, lines 45/46: please change to 'help ensure'

We have made this suggested change (page 39, line 16).

Reviewer: 2

I thank the authors for responding to the questions/comments previously raised

\

VERSION 3 – REVIEW

REVIEWER	Richard Keers The University of Manchester and Greater Manchester Mental Health NHS Foundation Trust, United Kingdom
-----------------	--

	MORPh Consultancy Ltd: received honorarium for speaker role at Primary Care Pharmacist Event: Mental Health (October 2019). Centre for Postgraduate Pharmacy Education (CPPE): received honorarium for development and maintenance of national learning materials. College of Mental Health Pharmacy (CMHP - registered charity no: 1141467): Council member
REVIEW RETURNED	30-Oct-2019

GENERAL COMMENTS	All of my comments have now been adequately addressed. The authors have asked the editor to review my recommendation to consider collapsing down categories of ethnicity with n=1 participants to avoid any risks of identifying individuals who took part. I agree that the editor should have final judgment on this issue. One possible solution is to collapse the 'Black' and 'Asian' categories together and keep 'Other' and 'White' categories separate, although one category of n=1 would still remain.
---